# Modelling an Optimal Climate-Driven Malaria Transmission Control Strategy to Optimise the Management of Malaria in Mberengwa District, Zimbabwe: A Multi-Method Study Protocol

**DOI:** 10.3390/ijerph22040591

**Published:** 2025-04-09

**Authors:** Tafadzwa Chivasa, Mlamuli Dhlamini, Auther Maviza, Wilfred Njabulo Nunu, Joyce Tsoka-Gwegweni

**Affiliations:** 1Department of Environmental Health, Faculty of Environmental Science, National University of Science and Technology, Bulawayo 00263, Zimbabwe; tafftaffie@gmail.com; 2Department of Environmental Health Services, Mberengwa District Hospital, Ministry of Health and Child Care, Mberengwa 00263, Zimbabwe; 3Department of Applied Mathematics, Faculty of Applied Science, National University of Science and Technology, Bulawayo 00263, Zimbabwe; mlangeni2007@gmail.com; 4Department of Environmental Science, Faculty of Environmental Science, National University of Science and Technology, Bulawayo 00263, Zimbabwe; mavizaa@gmail.com; 5Global Change Institute, Faculty of Science, University of Witwatersrand, Johannesburg 2050, South Africa; 6Department of Environmental Health, School of Public Health, Faculty of Health Sciences, University of Botswana, Gaborone 00267, Botswana; 7Department of Public Health, Faculty of Health Sciences, University of the Free State, Bloemfontein 0027, South Africa; tsokagwegwenijm@ufs.ac.za

**Keywords:** climate change, malaria transmission, management, Geographic Information System, Mberengwa, Zimbabwe

## Abstract

Malaria is a persistent public health problem, particularly in sub-Saharan Africa where its transmission is intricately linked to climatic factors. Climate change threatens malaria elimination efforts in limited resource settings, such as in the Mberengwa district. However, the role of climate change in malaria transmission and management has not been adequately quantified to inform interventions. This protocol employs a multi-method quantitative study design in four steps, starting with a scoping review of the literature, followed by a multi-method quantitative approach using geospatial analysis, a quantitative survey, and the development of a predictive Susceptible-Exposed-Infected-Recovered-Susceptible-Geographic Information System model to explore the link between climate change and malaria transmission in the Mberengwa district. Geospatial overlay, Getis–Ord Gi* spatial autocorrelation, and spatial linear regression will be applied to climate (temperature, rainfall, and humidity), environmental (Land Use–Land Cover, elevations, proximity to water bodies, and Normalised Difference Vegetation Index), and socio-economic (Poverty Levels and Population Density) data to provide a comprehensive understanding of the spatial distribution of malaria in Mberengwa District. The predictive model will utilise historical data from two decades (2003–2023) to simulate near- and mid-century malaria transmission patterns. The findings of this study will be used to inform policies and optimise the management of malaria in the context of climate change.

## 1. Introduction

Malaria remains a disease of public health concern. According to the 2022 World Malaria Report, there are approximately 241 million malaria cases and 627,000 deaths worldwide [1]. The report highlighted climate change as a key issue threatening progress in achieving set goals in the fight against malaria [2]. Malaria is a significant public health burden in Zimbabwe, ranking as one of the top five causes of morbidity, accounting for 10% of outpatient visits and 4% of deaths in the country [3]. More than half of the people in Zimbabwe live in areas at risk for developing malaria [4]. Children under the age of five years and pregnant women bear the highest burden of getting sick or dying from the disease [5,6]. Malaria in Zimbabwe leads to economic losses, reduced productivity, and increased strain in resource healthcare systems. A study conducted in Zimbabwe in 2014 revealed that malaria costs approximately $3.2 billion per year, which translates to 4.4% of the country’s GDP [7].

Climate factors, particularly temperature, rainfall, and relative humidity, are among the key drivers of malaria transmission dynamics in the Mberengwa district and in similar settings [8,9,10]. Studies have shown that they influence the lifecycle and abundance of both the parasite and its primary vector, Anopheles mosquitoes [11]. Malaria parasite development and vector mosquito survival require a certain temperature range [12]. Some authors have suggested that at temperatures above 32 degrees Celsius (°C), mosquito vector turnover is high at temperatures above 32 °C, but mortality increases with temperature [13,14]. Warmer temperatures can extend the malaria season by creating more suitable malaria habitats and increasing the vector mosquito populations. Rainfall also plays a fundamental role in creating potential breeding sites for vector mosquitoes that are known to breed in stagnant waters [9]. Similarly, relative humidity is also a driving factor for malaria transmission, considering that the period required to complete the lifecycle of mosquitoes is influenced by the amount of moisture in the atmosphere. Given the complexity of environmental influences, optimal control theory has become a valuable tool in malaria management. This theory can be useful in providing a structured framework for selecting and dynamically adjusting intervention strategies based on specific objectives and resource availability, thereby ensuring that control measures are both effective and sustainable [15].

The successful implementation of vector control strategies in malaria management is only possible when there is adequate knowledge of the malaria transmission dynamics. Many epidemiologists have invested considerable effort and time into understanding these dynamics and suggesting suitable control measures. This knowledge has been used to develop models and to provide insights into host and vector populations [16,17]. Models are crucial for malaria research and the development of optimal control and elimination strategies, informing the choice of areas to target, which interventions to consider, and which risk groups to focus on. Spatial epidemiology, which combines mathematical modelling and geographical information systems (GIS), is becoming increasingly mainstream in the field of malaria transmission, as it allows researchers to explore the intricate relationships between environmental, socio-economic, and biological factors that influence the transmission of malaria [18,19,20]. Epidemiologists have used spatial autocorrelation measures to identify and quantify hotspots for infectious diseases such as malaria. Spatial autocorrelation refers to the tendency of a variable to correlate with itself in space, and it is crucial for understanding the clustering of diseases in space. This study integrates this key concept to enhance the understanding of malaria case distribution in Mberengwa District.

In Zimbabwe, malaria transmission differs across districts, with areas such as Chiredzi, Mwenezi, and Binga still experiencing an annual parasite index (API) above 5 per 1000 people, necessitating ongoing malaria control measures. In contrast, districts such as Mberengwa, Gokwe South, Gwanda, and Tsholotsho, where the API has dropped below 5 per 1000 people, have transitioned to pre-elimination activities [3]. The transition from malaria control to elimination in districts such as Mberengwa makes them priority areas for this type of study, offering insights into the threats of climate change to current interventions and the challenges of sustaining low transmission rates. Like other parts of Zimbabwe, Mberengwa may experience shifts in malaria transmission due to climate change. Selecting this district allows for a focused study of how climate factors influence transmission patterns and the effectiveness of control strategies in a pre-elimination setting.

Mberengwa District was enrolled in the malaria elimination phase in 2018 after decades of control intervention [5]. Malaria elimination aims to create an environment in which communities are free from local malaria transmission, ensuring that residents can live and work without the burden of this life-threatening disease [21]. Over the years, the district health office has implemented various control measures, including the distribution of insecticide-treated bed nets, indoor residual spraying, environmental manipulation, larval source management, and communication of social behaviour change with international partners. Despite these efforts, the Mberengwa district has continued to experience localised outbreaks not only in traditional malaria areas but also in new regions which historically did not report cases, highlighting the potential changing dynamics and increased risk of losing the elimination gains achieved over the years [22]. This protocol introduces a novel approach by integrating GIS and mathematical modelling to identify malaria hotspots and predict future malaria peaks in the Mberengwa district, which is not currently employed in existing studies that focus on malaria and climate relationships. Few studies have utilised robust multi-method approaches to inform climate-sensitive malaria control interventions in limited resource settings [23]. There are several other protocols in literature which have attempted to address the issue of malaria transmission and climate change, including the Liverpool Malaria Model [17,24], but they often lack integration of behavioural, socio-economic, and climate-driven factors. Our approach is more comprehensive and enhances adaptability to local settings, making it more robust compared to existing models.

The impact of malaria on the community is devastating, resulting in loss of productivity, school absenteeism, loss of lives, especially for children and pregnant women, and an economic burden on households, perpetuating the cycle of poverty in the face of persistent droughts [25]. Even in areas that report sporadic cases, the local rural population continues to bear the burden of limited access to healthcare and resources. Despite existing elimination efforts, the persistent nature of malaria in Mberengwa District necessitates this research, which aims to model an optimal climate-driven malaria transmission control strategy to optimise malaria management in the district. Although existing research has shed light on malaria control, there is a notable knowledge gap in the development of context-specific climate-informed strategies tailored to Mberengwa’s unique epidemiological and environmental profile, characterised by unstable patterns, variable climates, and limited resources [26]. To this end, this study seeks to leverage geospatial techniques to provide evidence-based solutions to strengthen malaria elimination efforts in the Mberengwa district, ultimately contributing to the attainment of the district’s development goals. The specific objectives of this study are presented in Table A1.

## 2. Materials and Methods

### 2.1. Research Approach

The study will be conducted in four steps to develop an optimal climate-driven malaria transmission control strategy for the Mberengwa district. The first step is a scoping review of the literature to synthesise historical and current knowledge on the impact of climate change on malaria transmission and management in Zimbabwe. The scoping review will reveal gaps, trends, key determinants, and key insights into the methods proposed in the literature to address the issues of malaria and climate change in similar settings. This step aims to provide a solid foundation for subsequent analysis by informing the selection of variables and validating the study findings. The next three steps apply a multi-method quantitative approach, leveraging geospatial analysis techniques in a geographical information system (GIS) environment, a quantitative survey, and the development of a predictive Susceptible-Exposed-Infected-Recovered-Susceptible Geographic Information System (SEIRS–GIS) model [27].

The use of multiple quantitative methods leverages the strengths of various quantitative techniques, providing a basis for a more comprehensive understanding of malaria transmission dynamics, including spatial patterns, environmental factors, and human behaviour [28]. The choice of GIS analysis is meant to provide visual insight into the spatio-temporal distribution of malaria cases relative to climate, environmental, and socio-economic factors. It will also enable the identification of areas with a high potential for transmission, informing the prioritisation of key interventions. The GIS findings will be integrated with findings from the quantitative survey to enhance the understanding of how local environmental and socio-economic factors influence malaria occurrence in elimination settings. A quantitative survey is included to capture the human dimension, in addition to the climatic and environmental insights from the GIS and SEIRS–GIS models. It provides context-specific evidence of individual behaviours and socio-economic and structural challenges. Socio-economic, individual, and structural determinants analysed in the quantitative survey could not be captured by the other methods.

The integration of findings from the survey and GIS analysis will cover both the spatial and non-spatial determinants of malaria transmission and management, providing reliable evidence for equitable interventions. The inclusion of a SEIRS–GIS model in this study is meant to inform current and future intervention strategies by identifying future transmission patterns and hotspots. Predictive analysis will guide the planning of future malaria elimination activities. GIS analysis is descriptive in nature. The combination of the SEIRS mathematical model and GIS provides a platform for spatially sensitive predictions, making it possible to predict potential geographical shifts in malaria transmission under different climate change scenarios. This multi-method approach considers the complex interlinkages among environmental, climatic, socio-economic, and human factors, providing a holistic understanding of malaria transmission and informing the development of effective control strategies.

### 2.2. Study Area

The study will be conducted in the Mberengwa district (Appendix A). The district was purposely selected because of its high vulnerability to climate change, persistent malaria outbreaks, and unique socio-environmental conditions which make it ideal for developing and testing an optimal climate-driven malaria transmission control strategy. It lies in natural region IV, which is characterised by low rainfall (500–700 mm) and high temperatures [29]. This semi-arid climate is particularly sensitive to climate fluctuations, making it susceptible to droughts, floods, and infectious disease outbreaks (e.g., cholera and malaria). The occurrence of extreme weather events, such as El Niño, can potentially affect mosquito breeding patterns and malaria transmission dynamics [30]. Mberengwa’s varied topography, encompassing mountains, valleys, and plains, creates diverse microclimates that influence mosquito breeding and human–vector interactions [31]. This heterogeneity provides an opportunity to include the spatial dimension in analysing malaria–climate interlinkages and tailor-make interventions. The warm and humid climate that characterises the district makes it ideal for malaria vector mosquito breeding. The Mberengwa district has seen fluctuations in malaria incidence from 2018 to 2024, with a notable increase in 2020, followed by a decline in 2021 and 2022. In 2023, there was a sudden increase in malaria cases, suggesting the need for vigilance, especially given the potential impact of climate change on malaria transmission patterns [22]. The instability in malaria transmission dynamics has threatened the gains made in reducing the malaria burden in the district over the past few decades.

### 2.3. Step 1: Scoping Review of Literature

#### 2.3.1. Review Title

Impact of climate change on malaria transmission and management in Zimbabwe: A scoping review of literature.

#### 2.3.2. Background of the Review

Malaria continues to exert persistent public health challenges, particularly in endemic regions such as Sub-Saharan Africa (SSA) [32]. According to the 2022 World Health Organisation Malaria Report, the region accounts for approximately 95% of global malaria cases and 96% of malaria deaths [2]. Zimbabwe is among the highly burdened countries in SSA, with rural communities disproportionately affected due to poor access to health services and limited infrastructure, among other individual and structural factors [32]. Despite the orientation towards malaria elimination, the disease remains endemic and continues to claim lives, accounting for approximately 876 of deaths in 2022 [33].

Studies have revealed that the transmission of malaria is highly sensitive to climatic factors, particularly temperature, rainfall, and humidity, which influence vector and parasite lifecycles [2,30]. The Intergovernmental Panel on Climate Change (IPCC) highlighted that climate change might have profound effects on climate-sensitive vector-borne diseases, such as malaria [34]. In addition, climate change is anticipated to alter vector distribution and behaviour, leading to the extension of malaria seasons and active foci [10,23]. Climate change can result in extreme weather events, such as El Niño, which can create favourable environments for malaria transmission in previously non-receptive areas. However, extreme weather events can potentially reduce vector survival, further complicating the relationship between climate change and malaria transmission [35].

Climate change is likely to pose significant challenges to malaria management, especially in limited resource settings. Changes in vector distribution and behaviour have the potential to render current vector control strategies ineffective [36]. For instance, vector control in the form of Indoor Residual Spraying (IRS) and insecticide-treated nets (ITNs) may lose its effectiveness if vectors change their resting and biting behaviours or if vector populations are ready for new geographical areas [6]. Furthermore, climate change can strain Zimbabwe’s fragile health system by exerting greater pressure on its limited resources and infrastructure. The increased malaria burden on previously malaria-free populations may overwhelm health services, resulting in the late diagnosis and treatment of malaria cases [37]. The lack of adequate resources and infrastructure which are common in the country makes it difficult to adapt to the changing malaria epidemiological landscape [38].

Scholars are increasingly working to provide evidence of the impact of climate change on malaria; however, significant gaps exist in the understanding of how these changes specifically affect malaria transmission and management in Zimbabwe and similar settings [9,10,13]. Existing research has not provided adequate information on the actual changes occurring on the ground due to climate change. Moreover, existing research in Zimbabwe is fragmented, with studies focusing on small areas or specific aspects of the relationship between malaria and climate change [39,40,41,42]. Conducting a scoping review of the literature provides an opportunity to systematically address these gaps by mapping the existing literature, identifying key themes, and identifying areas for future research. This review will provide an overview of the current state of knowledge, analyse proposed strategies to optimise malaria management, and inform future research and policies to manage malaria in the country.

#### 2.3.3. Main Objective

This study aims to synthesise historical and current evidence on the impact of climate change on malaria transmission and management in Zimbabwe. The specific objectives of this study are summarised in Table A1.

#### 2.3.4. Methodology

##### Inclusion and Exclusion Criteria

This literature review will only focus on studies conducted in sub-Saharan Africa from January 2000 to the end of September 2024. This period may provide historical and modern insights into the impact of climate change on malaria transmission and management. The focus will be on grey literature and peer-reviewed research articles written in English and published in reputable databases with open access. The selection will include studies that assess measurable outcomes related to climate change and malaria transmission. The review will consider studies accessible from reputable databases, such as PubMed, Web of Science, African Journals Online, EBSCO, Cochrane Library, and ScienceDirect. Grey literature will include government and non-governmental organisation reports, policy briefs and documents, discussion articles, and conference papers. This scoping literature review will exclude studies that did not focus on malaria transmission or control, even if conducted in Zimbabwe. In addition, the review will exclude studies that are from non-peer-reviewed databases, research protocols, books, newspaper articles, and studies conducted in languages other than English or partially conducted in English. This approach streamlines the review process and covers relevant studies that will help inform effective strategies for addressing the impact of climate change on malaria transmission and management.

##### Search Strategy

The Google Advanced Search Engine will be used to access relevant studies from PubMed, Web of Science, African Journals Online, EBSCO, Cochrane Library, and ScienceDirect. The keywords for the systematic review are Climate change, Malaria Transmission and management, and Zimbabwe. Electronic databases will be searched for data published between January 2000 and September 2024. The following search string will be used: “climate change” OR “climate variability” OR “climate factors” OR “environmental factors” AND “malaria transmission” OR “malaria incidence” OR “malaria control” OR ”Malaria elimination” OR “malaria management” AND “rainfall” OR “temperature” OR “humidity” OR “landcover land use (LCLU)” OR “Normalised Difference Vegetation Index (NDVI)” OR “elevation” OR “proximity to water bodies” AND “Zimbabwe” AND “quantitative study” OR “quantitative analysis” AND “qualitative study” OR “qualitative analysis” AND “peer-reviewed” OR “research articles” AND “open access” AND (“2000/01/01”[Date—Publication]: “2024/09/30”[Date—Publication]) AND English[Language] to search for eligible literature sources in the selected databases. Snowballing will be used to search and identify relevant articles and websites.

##### Methods of Review

This review will implement a two-stage screening process that initially screens titles and abstracts according to inclusion and exclusion criteria [43]. The selected articles will then undergo a thorough examination of the full text to assess their alignment with the research question, methodological rigor, and quality of data analysis. Each article will be reviewed by at least two reviewers using Rayyan software (https://www.rayyan.ai/ accessed on 9 Mach 2025) to identify articles that would be relevant for the final consideration. Disagreements will be resolved through dialogue between the reviewers. If consensus is not reached between the two reviewers, a third reviewer will be consulted to give the final decision.

##### Data Extraction and Synthesis

Importation of all downloaded data into Rayyan software will be done to ensure that only relevant sources are included for analysis. Rayyan is a web and mobile app specifically designed to help researchers work on literature reviews more efficiently by ensuring that all work done on the subject matter is accessed to obtain comprehensive findings [44]. A data extraction form will be obtained from the JBI guideline manual, and a scoping review checklist from the manual will be used. At least two reviewers will participate in the extraction of key information from each included article. The key data elements to be included in the data extraction form were the evidence source, year of publication, study focus, methodology, methods applied, data collection methods, location, population, strategy effectiveness, climate impact on efficacy, proposed interventions, intervention effectiveness, and future projections. The analysis will focus on the core concepts of climate-driven malaria transmission and management strategies. Narrative synthesis and thematic analysis techniques will be used to identify key themes, patterns, and gaps in knowledge [45]. The process of collecting and analysing data, culminating in the submission of a final review paper to a journal, will take place between October and December 2024.

##### Quality Assessment

The quality of the included evidence will be assessed using the Mixed Methods Appraisal Tool (MMAT). MMAT is a highly suitable quality evaluation tool for a scoping review of the literature owing to its versatility and comprehensive criteria for different study designs [46]. Given that the review will capture evidence in various forms (qualitative, quantitative (descriptive and analytical), and mixed methods), MMAT is suitable for such a heterogeneous review. Methodological rigor will be assessed by analysing the study design, sampling, data collection, and analysis. Articles will be further assessed for clarity in presenting the relationship between climate change, selected (e.g., topographic and socio-economic) variables, malaria transmission, and existing management strategies.

### 2.4. Step 2: Geographic Information System

#### 2.4.1. Data Collection Methods and Tools

The GPS device will be used to capture coordinates for health facilities and upload them to QGIS for the analysis and visualisation of spatial data. Climate data (temperature and humidity) for the period 2003–2023 will be obtained from the European Centre for Medium-Range Weather Forecasts Reanalysis 5th Generation (ECMWF–ERA5). Rainfall data for the same period will be acquired from the Climate Hazards Group Infrared Precipitation with Station (CHRPS) Dataset. Remote sensing data for environmental (Land Use-Land Cover (LULC), Elevation, Proximity to water bodies, and Normalised Difference Vegetation Index (NDVI)) variables will be accessed from Landsat 8 OLI/TIRS imagery from the U.S. Geological Survey (USGS) and MODIS provided by NASA’s LP DAAC. Socio-economic variables (poverty levels and Population Density) will be accessed from the Zimbabwe Census and Zimbabwe Vulnerability Assessment Committee (ZIMVAC) reports. Data for malaria cases will be collected from the Ministry of Health and Child Care District Health Information Software 2 (DHIS2) for the period 2003–2023. The period from 2003 to 2023 allows adequate time for significant climate fluctuations, intervention efforts, and policy changes in malaria management. Additionally, the two-decade span will provide sufficient data for model accuracy validation and prediction of future malaria transmission patterns [47,48,49].

#### 2.4.2. Data Management, Analysis and Presentation

Climate and environmental data will be aggregated by year for each of the 37 health facilities for 2003–2023. The malaria incidence data will be merged with the corresponding climate and environmental data for each health facility by year. R software 4.5.0 packages will be used for data extraction, cleaning, and processing. The integrated dataset will be standardised to ensure consistency in format and units. Climate variables, such as temperature, rainfall, and humidity, often have different units and ranges; therefore, z-score normalisation will be applied to allow comparison without the influence of their original units. Varying ranges of malaria incidence data and socio-economic variables will be transformed into a uniform scale by applying min–max scaling to facilitate comparison and integration into models. Sudden outbreaks are common during peak malaria seasons, which can result in skewed data [48]. Therefore, this study will apply log transformation to such data to make patterns more interpretable. Since it is noted that rainfall can influence malaria incidence with lag time, which is the period required for mosquito populations to respond to changes in rainfall patterns, all statistical analyses will incorporate lagged rainfall data [50]. This will capture the delayed effects of rainfall on vector mosquito breeding and its subsequent influence on malaria transmission. All captured and/or pre-processed geodata sets (satellite imagery, data products, and vector files) will be assigned the same coordinate reference system for easy overlay in the GIS. QGIS software 3.42.0 will be used to overlay malaria incidence, climate, and environmental variables to visualise their spatial relationships and patterns. Exploratory Data Analysis will be performed to understand the distribution and trends of malaria incidence, climate, and environmental variables. Trend analysis will be applied to understand temporal changes in malaria incidence and related climate and environmental variables to obtain insights into how these factors have evolved and their potential correlations.

Spatial autocorrelation analysis will be conducted to identify the relationships between malaria incidence and the selected (climate and environmental) variables. The study will employ Moran’s I to measure spatial autocorrelation and detect malaria hotspots in the Mberengwa district. Spatial autocorrelation analysis will be complemented using the Getis–Ord* Statistic to identify statistically significant malaria hotspots. Spatial linear regression will be used in the analysis to obtain a comprehensive understanding of the spatial distribution of malaria. Multiple map panels will be generated using GIS to visualise the geographic distribution of malaria incidence and hotspots identified by Getis–Ord* analysis. Heat maps and choropleth maps will be developed to illustrate the spatial interlinkages between climatic and environmental conditions and malaria incidence.

### 2.5. Step 3: Quantitative Survey

#### 2.5.1. Data Collection Methods and Tools

The quantitative survey will use secondary data from case-based surveillance activities to determine the driving factors of malaria transmission. Three data collection forms (malaria case notification, malaria case investigation, and malaria foci classification) will be used to collect the data. Data will be accessed from the District Health Information Software-Tracker through the Ministry of Health and Child Care. Malaria case-based surveillance questionnaires were first introduced in the Mberengwa district in 2018, when the district began conducting malaria-enhanced surveillance activities. In this study, malaria-positive cases (RDT+ or Slide+) reported in the Mberengwa district from 2019 to 2024 will be considered in the analysis. The current study considers cases beginning in 2019 when data quality improved, when health workers gained more experience in using the data collection tools and data quality had improved. Retrospective convenience sampling will be used to collect data for the study. In this case, the existing data collected from January 2019 to December 2023 will be used. During the four years, the district reported 846 cases, with a case investigation rate of 44% [51]. The study will use (374) malaria cases that were fully investigated and synchronised through the DHIS-Tracker. The selection of this complete and high-quality data subset provides a feasible and reliable sample for this research.

#### 2.5.2. Data Management, Analysis and Presentation

Information on case-based surveillance data will be accessed in Excel format from DHIS-2. The R software package will be used to clean and validate the data, which will be securely stored in a password-protected database. Descriptive statistics will be used to summarise the demographic and clinical characteristics of the malaria cases. The mean and median will be used as measures of central tendency for key malaria-driving factors. The dependent variable is malaria incidence, whereas the independent variables are structural, behavioural, and socio-economic factors. Bivariate analysis will be used to compare relationships and identify the driving factors associated with malaria. Multivariate logistic regression analysis will be used to identify the structural and individual determinants of malaria transmission in the Mberengwa district. Model fit will be evaluated using the Hosmer–Lemeshow test. Odds ratios (OR) and 95% confidence intervals (CI) will be calculated for each independent variable to understand the strength and direction of the associations. For all statistical analyses, the Statistical Package for Social Sciences (IBM SPSS 29.0.2.0(20)) for Windows will be used.

### 2.6. Step 4: Predictive SEIRS–GIS Model

#### 2.6.1. Data Collection Methods and Tools

The entire population residing in the Mberengwa district, which is divided into 37 administrative wards serviced by 37 health facilities, will be considered in this model. This population forms the basis for understanding malaria transmission patterns and for evaluating the effectiveness of control measures. Historical climate, environmental, intervention, and malaria incidence data for the period from January 2003 to December 2023 will be considered in this study. Climate data for 2003–2023 will be obtained from the European Centre for Medium-Range Weather Forecasts–Reanalysis 5th Generation (ECMWF-ERA5) and Climate Hazards Group Infrared Precipitation with Station Data (CHRPS) databases. Remote sensing data for environmental variables will be accessed from Landsat 8 OLI/TIRS imagery from the United States Geological Survey (USGS) and MODIS, provided by NASA’s LP DAAC. Malaria incidence data will be collected from the Ministry of Health and Child Care District Health Information System Software (DHIS2). Incomplete and inconsistent data will be managed through K-nearest neighbour (K-NN) imputation by estimating missing values based on observed patterns. A multiplication factor from a scoping review of the literature will be used to adjust for under-reporting. Kernel density estimation will be applied to account for spatial variations in malaria reporting rates.

#### 2.6.2. SEIRS Model Building and GIS Integration

The SEIRS model compartments are defined as follows: susceptible (S), exposed (E), infectious (I), recovered (R), and susceptible (S). Infection rate, recovery rate, immunity duration, birth rate, death rate, and climate variables were incorporated into the model. Epidemiological and climate data will be used to estimate model parameters. The maximum likelihood estimation statistical method will be applied for the parameter estimation. QGIS software will be used to map and develop the requisite spatial layers for malaria incidence, climate, and environmental variables for input into the SEIRS model. Spatial layer integration with the SEIRS model will be performed to account for geographic variability in malaria transmission dynamics.

The study will use projections from two climate change scenarios, SSP2-4.5 (moderate scenario with gradual climate action and socio-economic development) and SSP3-7.0 (pessimistic pathway characterised by regional rivalry, limited mitigation, and higher emissions) to predict future climate impact on malaria transmission dynamics. SSP2-4.5 and SSP3-7.0 are part of the framework for Shared Socio-economic pathways (SSPs). They help predict future climate conditions through the integration of socio-economic narratives and Representative Concentration Pathways (RCPs) [52]. SSP2-4.5 reflects a world with some progress in reducing emissions with no transformative changes, whereas SSP3-7.0 is a high emission with limited mitigation efforts. The major assumptions for SSP2-4.5 include intermediate population growth, balanced energy use with the reduced use of fossil fuels, and moderate economic growth [53]. The SSP3-7.0 assumes a world with regional conflicts and weak international corporations and focuses on national interests. Both scenarios assume that malaria transmission is directly influenced by temperature and precipitation, and their projections are based on mid-century climate data [54]. The projected climate data will be incorporated into the SEIRS–GIS model to simulate future malaria transmission, enabling a comprehensive understanding of how varying climatic conditions influence malaria risk.

#### 2.6.3. Model Validation and Sensitivity Analysis

The SEIRS–GIS model will be validated using a split-sample approach for the 2003–2023 dataset. The data will be divided into two parts: a training dataset (2003–2015) for parameter estimation, and a validation dataset (2016–2023) for testing the model’s predictive ability on unseen malaria incidence and climate change data. Model predictions will be compared with observed data to assess accuracy. Root Mean Squared Error (RMSE), Akaike Information Criterion (AIC), and Bayesian Information Criterion (BIC) were incorporated into the model to assess its accuracy and calibration. In cases where multiple models demonstrate comparable performance, the model averaging approach will be explored to leverage the combined strengths of the individual models. Sensitivity analysis will be performed to identify the key parameters influencing model outcomes, and the robustness of model predictions will be assessed under different parameter settings. Key epidemiological variables (incidence rate and recovery rate) and climatic factors (e.g., temperature, rainfall, and relative humidity) will be selected for sensitivity analysis. The choice of climate variables to be included in the sensitivity will be determined by their level of influence on malaria transmission in Mberengwa District, as informed by findings from the scoping review and GIS analysis.

#### 2.6.4. Predictive Analysis

A simulation of different malaria control strategies will be conducted (e.g., increased ITNs coverage, increased IRS coverage, and combined IRS and ITNs). The study will focus on near-future projections (2021–2040) to gain insights into short-term malaria risks and the mid-century period (2041–2060), where climate impacts on malaria transmission are anticipated to be more pronounced. The model will also be used to identify future high-risk areas for targeted interventions. Predictive simulations under different climate scenarios (SSP2-4.5 and SSP3-7.0) will be conducted using R software.

#### 2.6.5. Reliability and Validity

Sensitivity analysis will be conducted to assess how the model parameters and assumptions affected the model outputs. In addition, a sensitivity analysis will be employed to assess the impact of data quality issues on the outcome. The GIS and mathematical model outputs will be validated using observed data from the Mberengwa district. Multiple epidemiological data sources will be used to confirm and fill in missing data, including DHIS-2, malaria case investigation reports, and health facility registers. The data will be collected, cleaned using Microsoft excel Excel 2019 and Python version 3.10, and checked for completeness and consistency before analysis in the relevant software. The research protocol was reviewed and approved by the National University of Science and Technology Institutional Review Board (NUST IRB). Both the methodology and the data collection instruments were adjusted according to the input received.

### 2.7. Pilot Testing

Pilot testing will be conducted to identify logistical, methodological, and data-related challenges before full-scale implementation. This will be conducted using a structured approach focusing on data collection tools, GIS workflows, and model parameters. The feasibility and reliability of the data collection instruments will be assessed at this stage. To assess the success of the pilot test, the key performance indicators to be employed will include data completeness, consistency, and time required for data collection and processing. The pilot test will also focus on testing and refining SEIRS model parameters through running preliminary simulations with a subset of data. This will help to identify issues with parameter sensitivity and model assumptions. The ability of the model to replicate known malaria trends in Mberengwa District will be used as a measure of the success of the text. Additionally, the results should remain stable under various input scenarios. Parameters such as transmission rates and environmental factors will be adjusted based on insights from this test. In the case of GIS workflows, the efficiency and accuracy of handling multiple layers of data, such as climate data, malaria incidence, and poverty levels, will be assessed. The success of this stage will be measured by the spatial resolution of the map outputs and processing time. GIS methodologies will be adjusted based on the results of the pilot test to improve the accuracy of spatial analysis and interoperability. Overall, pilot testing will minimise disruptions through a proactive approach that identifies challenges and ensures a smooth full-scale implementation [55].

### 2.8. Informed Consent

The data used in this study will be anonymised and aggregated; therefore, no additional informed consent will be required for this secondary analysis. The protocol was approved by the NUST Institutional Review Board to ensure that the research was ethical and the rights of the participants are protected.

### 2.9. Data Handling

The collected data will be electronically stored on a secure password-protected device, with access restricted to the research team. To prevent unauthorised access, the data will be de-identified and encrypted. There will also be regular backing of data, which will be created and stored securely. Secure File Transfer Protocols (SFTP) will be used to transfer data between systems to prevent unauthorised access during the process. The data will not be shared with any third parties or used for purposes other than this study. All sensitive data, including backups, will be retained for a five-year period post-publication and permanently deleted thereafter.

### 2.10. Risks to Participants

There will be no direct risk to participants because the data have already been collected and anonymised, ensuring participants’ privacy and confidentiality.

### 2.11. Benefits to Participants

Although the current study used secondary data and did not involve direct participation, its findings have the potential to enormously benefit the Mberengwa community and beyond in their endeavour to eliminate malaria. The findings of this study will help climate-proof future malaria elimination strategies and sustain the gains achieved over the past decades to eliminate malaria in Mberengwa District.

### 2.12. Confidentiality

The data for case-based surveillance, malaria epidemiology, and climate variables will be obtained from reputable organisations and anonymised to protect participants’ privacy. A rigorous anonymisation process will include the removal of all personal identifiers, such as names and addresses, that can directly or indirectly link data to individual participants. Furthermore, unique identifiers from the datasets will be replaced with randomly generated codes to protect the confidentiality of participants during data analysis. Access to data is restricted to the research team, and research findings will be presented in a manner that maintains participant confidentiality and anonymity.

### 2.13. Knowledge Dissemination

The study results will be shared with key stakeholders through workshops for government officials, policy briefs tailored to health organisations, and simplified presentations for non-specialist audiences. Oral presentations will be used to communicate the findings at national and international conferences. The findings of this study will be published in peer-reviewed journals that focus on epidemiology, public health, and climate change. Communities will be engaged in participatory meetings to advocate the adoption of the proposed interventions into local health strategies.

## 3. Discussion

Studies that quantify the specific relationships between malaria transmission and established climate, topographic, and socio-economic factors in malaria elimination settings, such as the Mberengwa district, are scarce. Furthermore, no known studies have captured the influence of microclimatic variations on malaria transmission in the Mberengwa District. This study aims to analyse malaria transmission dynamics in Mberengwa District, Zimbabwe, using a quantitative framework to assess the impacts of climate change and environmental factors on malaria transmission and management. A four-step multi-method quantitative approach will be employed to provide a deeper understanding of the complex dynamics of malaria transmission in the wake of climate change. A scoping review of the literature in Zimbabwe will lay the foundation for understanding the relationship between climate factors and malaria transmission in limited resource settings, including the Mberengwa district. Insights from this review will help to inform the status of adaptation to the impacts of climate change. These findings can help inform policies on the need to integrate climate adaptation into the National Malaria Control Program and the development of training materials for health workers involved in malaria control activities. A structured quantitative survey will utilise existing case-based surveillance data to gather empirical evidence on other key determinants of malaria transmission in Mberengwa District. The findings will highlight behavioural, structural, and socio-economic determinants and how they interact to influence malaria transmission. These insights will guide authorities in implementing interventions based on contextual evidence. For example, knowledge of determinants of malaria transmission will guide the designing of targeted Social Behaviour Change (SBC) programs to increase uptake of preventive strategies

The application of GIS techniques to spatially analyse the relationship between malaria incidence and climate (temperature, rainfall, and humidity), socio-economic (poverty levels and population density), and environmental variables such as proximity to water bodies, NDVI, and LULC will help in the identification of malaria hotspots and trends. The resulting heat maps and hotspot analysis will provide evidence of spatial variations in malaria incidence linked to environmental, socio-economic, and climatic factors. Recommendations will be on guiding resource allocation and development of climate-sensitive early warning systems. The last step will be the development of a predictive SEIRS–GIS model that will utilise historical data from two decades (2003–2023) to simulate near- and mid-century malaria transmission patterns. The model projections under SSP2-4.5 and SSP3-7.0 scenarios will have an implication on developing climate-resilient strategies to reduce malaria transmission. These strategies can include context-specific integrated vector control and climate-resilient infrastructure that reduces human–vector mosquito interactions. Ultimately, this multi-method approach provides a good platform for tailoring malaria control efforts in specific geographical settings [56,57].

The study will rely on DHIS2 for malaria incidence data, whereas climate and environmental data will be sourced from online databases. A SEIRS–GIS model will use insights from other quantitative approaches to identify optimal control strategies and predict future malaria transmission dynamics. The optimal control strategies developed would be suitable for a unique Mberengwa climate and community. The application of mathematical modelling to epidemiological and meteorological data adds value to data that has been collected for several years without being utilised to inform public health policies. Adapting an SEIRS mathematical model that captures climate variability, optimal control, and the use of GIS to map the risk factors for malaria transmission in the area will be a unique way to identify areas that are most at risk and predict future malaria peaks to inform interventions. The combination of mathematics, GIS, and epidemiology improves the robustness of this study by addressing malaria and climate change problems from multiple perspectives [39,56]. A scoping review of the historical and current literature will help interpret the study findings within the broader context of Zimbabwe. Evidence derived from this study is expected to inform current and future hotspots in Mberengwa District, uncover the nexus between specific climate and environmental variables, and provide predictive insights that will inform malaria management under climate change scenarios.

A collaborative approach involving engagement with key stakeholders will enhance data interpretation and facilitate the development of locally applicable malaria elimination strategies. Key stakeholders to be engaged throughout this study include community leaders at full council meetings, health officials during workshops, and partners involved in malaria programming. This will ultimately facilitate the adoption of evidence-based interventions based on study findings. Throughout the study process, the ethical implications of data reporting choices will be considered to prevent community stigmatisation. The reputation of affected communities will be protected by reporting results at the health facility catchment area level rather than pinpointing specific villages or wards and by providing contextual background to identified determinants. This will shift the focus from individual communities to prioritise action points to reduce malaria transmission in the context of climate change.

## 4. Limitations of the Study

While this protocol has been developed using validated robust methodologies, providing a systematic framework that can be used as required, certain limitations must be considered to balance the interpretation of the study findings. Contingency plans are provided to address the limitations of mitigating their impact on the results. The first major limitation is that the study lacks prior implementation to confirm its feasibility in a real-world environment. The uncertainty introduced by this limitation will be addressed by conducting a pilot study that focuses on a subset of the Mberengwa population before full-scale implementation. The logistical, data quality, and methodological gaps identified during the test bed pilot study will be used for appropriate refinement of the methodology. We propose that future research should focus on testing this protocol in different epidemiological contexts to validate its applicability. Second, incomplete data are common in retrospective epidemiological and meteorological datasets [58]. To mitigate this gap, data imputation techniques will be applied without compromising the integrity of the data. Third, there is inherent uncertainty in climate projections (SSP2-4.5 and SSP3-7.0) owing to variations in climate models and emission scenarios [54]. This will be addressed by performing a sensitivity analysis to assess its impact on the study findings.

Mathematical models may fail to adequately consider key variables, such as the effect of eventual vector control operations or public awareness. This gap will be mitigated using a multi-method approach, including a context-specific quantitative survey. Furthermore, the development of the SEIRS model is based on assumptions, such as population homogeneity, which may not portray a real-world scenario. Testing with real-world data from Mberengwa District will be conducted to address this limitation. Another additional challenge is the issue of data quality and consistency, resulting in variations in data collection methods over time. To address this concern, data processing protocols will be standardised and bias correction techniques will be applied. The validation of the historical data with official reports will be performed. Finally, logistical challenges pose yet another threat to the implementation of the study, and pilot testing will provide an opportunity to proactively identify and address gaps.

## Data Availability

Data are contained within the article.

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
