# Peer review of "Modelling an Optimal Climate-Driven Malaria Transmission Control Strategy to Optimise the Management of Malaria in Mberengwa District, Zimbabwe: A Multi-Method Study Protocol"

_ijerph, 2025, doi:10.3390/ijerph22040591_

Round 1
Reviewer 1 Report
Comments and Suggestions for Authors
The study protocol presents a robust framework that addresses the intersection of malaria control and climate change, an essential but underexplored area, particularly in pre-elimination regions such as Mberengwa District. It employs a comprehensive, multi-method approach by combining scoping reviews, geospatial GIS analysis, quantitative surveys, and predictive SEIRS-GIS models, enhancing the understanding of malaria dynamics. This interdisciplinary integration—merging epidemiology, climate science, and mathematical modeling—significantly strengthens the reliability of predictions and insights.
The study leverages two decades (2003–2023) of climate and malaria incidence data, ensuring that the models reflect real-world trends and allow for practical policy recommendations. The application of spatial autocorrelation techniques, such as Getis-Ord Gi* analysis within the GIS framework, offers precise identification of malaria hotspots, supporting targeted interventions. By focusing on Mberengwa District, the study tailors solutions to local conditions, enhancing the relevance and sustainability of its malaria control strategies. Additionally, the SEIRS-GIS model integrates different climate scenarios (SSP2-4.5 and SSP3-7.0), providing both short-term and mid-century projections to guide future interventions effectively.
While the protocol demonstrates substantial strengths, several aspects need further attention to improve clarity, coherence, and practical utility. Below are key areas of concern and suggestions for enhancing the study’s design and presentation:
1. The scoping review, GIS analysis, quantitative survey, and SEIRS model development contain overlapping elements, particularly in their focus on environmental factors influencing malaria transmission. For example, the scoping review synthesizes historical and current knowledge on how climate change affects malaria transmission and management in SSA. Meanwhile, the GIS analysis investigates relationships between climate variables (e.g., temperature, rainfall, vegetation) and malaria incidence to identify hotspots. The SEIRS-GIS model further incorporates epidemiological and environmental data to predict future transmission patterns, overlapping with the GIS analysis. Similarly, the quantitative survey explores drivers of malaria transmission through case-based surveillance, potentially duplicating insights from both the GIS analysis and scoping review. These overlaps create redundancy, which may disrupt the study's coherence. Refining the objectives for each component—scoping review, GIS analysis, quantitative survey, and SEIRS model—so that they align with specific research questions will help ensure that every step serves a distinct purpose and contributes to a logical and seamless narrative.
2. The manuscript mentions split-sample validation and sensitivity analysis for the SEIRS-GIS model but offers limited details on performance evaluation. A more rigorous framework is needed to ensure the model’s predictive reliability. Incorporating established metrics such as Root Mean Squared Error (RMSE), Akaike Information Criterion (AIC), or Bayesian Information Criterion (BIC) will help assess the model’s accuracy and calibration. If multiple models show comparable performance, model averaging could be applied to combine predictions and reduce uncertainty, enhancing predictive robustness. Additionally, specifying which variables will be used in the sensitivity analysis and providing a rationale for their inclusion would clarify how key parameters influence outcomes, making the model evaluation more transparent and robust.
3. Variability in data quality from 2018 to 2024 may impact the accuracy of the SEIRS-GIS model’s predictions. Missing values, inconsistent data entry, or gaps in surveillance coverage could introduce bias or weaken the model’s predictive power. To address these issues, the study should detail strategies for managing incomplete or inconsistent data during model development. Techniques such as multiple imputation or k-nearest neighbors (k-NN) imputation can estimate missing values based on observed patterns. Bias corrections, like adjusting for underreporting or weighting observations, will also improve data reliability. Additionally, sensitivity analysis can assess the impact of data quality issues on outcomes, ensuring that predictions remain robust across varying scenarios. These strategies will enhance the transparency and credibility of the study’s findings.
4. Greater detail is needed on how the study normalized data from 2003 to 2023 for integration across different models. The SEIRS-GIS model, GIS analysis, and quantitative survey rely on diverse datasets, including climate variables (e.g., temperature, rainfall) and epidemiological data (e.g., malaria incidence). Proper normalization ensures consistency in format, units, and scales, reducing bias and improving reliability. The manuscript should specify whether normalization methods such as z-score normalization, min-max scaling, or standardization were used and explain how these techniques were adapted for different variables. Transformations like logarithmic scaling might also be necessary to handle skewed distributions, particularly for malaria incidence data. Clarifying these details would enhance the reliability of correlations and predictions across the study’s components.
5. While the study relies on secondary data, the discussion on ethical considerations could be more comprehensive, especially regarding the anonymization and management of sensitive data. Since epidemiological data from the DHIS2 system may contain personal information, strict protocols are needed to ensure privacy and confidentiality. Additional detail on how personal identifiers will be removed or masked to prevent re-identification would strengthen the ethical framework. The manuscript should also describe measures to control access to sensitive data, such as using password-protected databases and encryption. Explicitly outlining these steps would demonstrate compliance with ethical standards and serve as a valuable resource for other researchers conducting similar studies by providing a framework for responsible data management.
6. Although the manuscript mentions pilot testing, it lacks detail on how the pilot phase will inform the full-scale implementation. Pilot testing is essential to identify logistical, methodological, or data-related challenges early in the process. The manuscript should specify criteria for evaluating the success of the pilot, such as key performance indicators or benchmarks, to determine whether the objectives were achieved. It would also be helpful to outline how insights from the pilot will inform adjustments to data collection tools, model parameters, or GIS workflows. Providing these details will enhance the transparency of the research process and ensure that the full-scale implementation is efficient, minimizing disruptions.
7. While the study acknowledges the role of human behavior in malaria transmission, it does not adequately address socioeconomic factors—such as income, education, healthcare access, housing, and occupation—that influence community vulnerability. These factors significantly impact healthcare-seeking behavior, adoption of preventive measures (e.g., bed nets), and adherence to malaria interventions. Integrating socioeconomic data is especially critical in regions like Mberengwa, where social inequities and limited resources can affect the effectiveness of control strategies. The study could explore how socioeconomic variables will be collected and integrated with environmental data to provide a more holistic understanding of transmission dynamics. For example, spatial overlays in the GIS analysis could include healthcare access or poverty levels to identify high-risk areas requiring targeted interventions. Examining interactions between socioeconomic conditions and behavioral factors—such as compliance with preventive measures—would provide actionable insights to inform equitable control strategies.
8. While the manuscript proposes visual tools such as heatmaps and choropleth maps to illustrate malaria transmission patterns, it lacks a clear plan for disseminating these findings to stakeholders. Effective communication is critical to ensure that policymakers, health officials, and other stakeholders can interpret and apply the results. Without a dissemination strategy, the study’s impact may be limited. Adding a section on knowledge translation would outline specific strategies for sharing the results. This could include workshops for government officials, policy briefs tailored to health organizations, interactive dashboards with real-time GIS data, or simplified reports for non-specialist audiences. Engaging local communities through participatory meetings or visual summaries would also enhance community buy-in and encourage the adoption of interventions. Ensuring that the findings are actively communicated will maximize their impact and integrate them into local health strategies.
9. While the manuscript employs multiple methods—GIS analysis, SEIRS-GIS modeling, and quantitative surveys—it does not provide sufficient clarity on why each was chosen or how they complement each other. Offering detailed reasoning for the inclusion of these methods will strengthen the study’s framework. For example, GIS analysis can identify malaria hotspots, the SEIRS model can predict future transmission trends under different climate scenarios, and quantitative surveys can capture behavioral and structural drivers of malaria transmission. Establishing these connections will ensure the methodology is well-coordinated, with each component contributing meaningfully to the study’s overall objectives.
10. The manuscript briefly mentions challenges but does not thoroughly explore potential limitations or contingency plans. Issues such as incomplete data, uncertainty in climate projections, and assumptions within the SEIRS model could affect the findings. A dedicated section discussing these limitations would improve transparency and help readers understand the study's scope. This section could address challenges like data gaps, forecasting inaccuracies, or logistical difficulties, and suggest contingency strategies such as alternative data sources, sensitivity analysis for varying assumptions, or pilot testing. Demonstrating preparedness will enhance confidence in the robustness of the research design.
11. Environmental variables like rainfall often influence malaria transmission with a lag, meaning that mosquito populations may increase weeks after a rainfall event. The manuscript does not address how such lag effects will be incorporated into the analysis. Including lagged variables in the statistical models will improve the accuracy of predictions by capturing these delayed impacts. For example, using lagged rainfall data in regression models can reveal delayed effects on mosquito breeding, providing a more realistic representation of the relationship between climate conditions and malaria incidence.
12. Although the study is policy-relevant, the manuscript does not specify how stakeholders—such as health officials or community leaders—will be engaged throughout the research process. Stakeholder involvement ensures the study aligns with local needs and increases the likelihood of successful implementation. It would be beneficial to describe how stakeholders will participate, such as through workshops to co-develop models, consultations to interpret results, or collaborative sessions to design intervention strategies. Early engagement will enhance the relevance and utility of the findings, ensuring they are translated into actionable outcomes.
13. While the manuscript discusses anonymization, it overlooks the potential risk of stigmatizing communities identified as malaria hotspots. If findings are not communicated sensitively, they could harm the reputation of affected areas. A responsible communication plan should be outlined to prevent such unintended consequences. Reporting results at a regional or district level rather than focusing on specific communities could reduce the risk of stigmatization. Collaborating with local representatives to interpret and disseminate the results will further promote ethical data use and foster community trust.
14. The manuscript mentions the use of SSP2-4.5 and SSP3-7.0 climate scenarios but does not provide enough background on these frameworks or their assumptions. A brief explanation of these scenarios would provide context, such as noting that SSP2-4.5 assumes moderate mitigation and socio-economic development, while SSP3-7.0 reflects limited mitigation with higher emissions. Clarifying any assumptions made in applying these scenarios to the SEIRS-GIS model will give readers a better understanding of the predictions’ strengths and limitations.
15. Although the manuscript focuses on visual outputs such as heat maps and predictions, it lacks practical guidance on how stakeholders should interpret and act on these findings. Without actionable recommendations, the insights generated may not be effectively utilized. Including examples of practical applications would enhance the study’s impact. For instance, if GIS analysis identifies high-risk areas, the study could recommend scaling up insecticide-treated net distribution or prioritizing indoor residual spraying. Providing clear guidelines on how policymakers and health officials can use the findings to allocate resources, design interventions, and monitor progress will ensure that the results are meaningful and actionable.
Author Response
|
1. The study protocol presents a robust framework that addresses the intersection of malaria control and climate change, an essential but underexplored area, particularly in pre-elimination regions such as Mberengwa District. It employs a comprehensive, multi-method approach by combining scoping reviews, geospatial GIS analysis, quantitative surveys, and predictive SEIRS-GIS models, enhancing the understanding of malaria dynamics. This interdisciplinary integration—merging epidemiology, climate science, and mathematical modeling—significantly strengthens the reliability of predictions and insights. |
We thank the reviewer for this positive feedback. We appreciate the recognition of our comprehensive multi-method approach and the interdisciplinary integration of epidemiology, climate science, and mathematical modeling. Our aim is to maintain this robustness throughout the study. |
|
2. The study leverages two decades (2003–2023) of climate and malaria incidence data, ensuring that the models reflect real-world trends and allow for practical policy recommendations. The application of spatial autocorrelation techniques, such as Getis-Ord Gi* analysis within the GIS framework, offers precise identification of malaria hotspots, supporting targeted interventions. By focusing on Mberengwa District, the study tailors solutions to local conditions, enhancing the relevance and sustainability of its malaria control strategies. Additionally, the SEIRS-GIS model integrates different climate scenarios (SSP2-4.5 and SSP3-7.0), providing both short-term and mid-century projections to guide future interventions effectively. |
We appreciate the detailed feedback. The use of spatial autocorrelation techniques, such as Getis-Ord Gi* analysis, was indeed intended to enhance the precision of hotspot identification. |
|
3. While the protocol demonstrates substantial strengths, several aspects need further attention to improve clarity, coherence, and practical utility. Below are key areas of concern and suggestions for enhancing the study’s design and presentation: |
We value your suggestions, we have taken note of the aspects and addressed them in the manuscript to improve clarity, coherence and practical utility. We therefore hope that the changes made further enhance the quality of the manuscript. |
|
4. The scoping review, GIS analysis, quantitative survey, and SEIRS model development contain overlapping elements, particularly in their focus on environmental factors influencing malaria transmission. For example, the scoping review synthesizes historical and current knowledge on how climate change affects malaria transmission and management in SSA. Meanwhile, the GIS analysis investigates relationships between climate variables (e.g., temperature, rainfall, vegetation) and malaria incidence to identify hotspots. The SEIRS-GIS model further incorporates epidemiological and environmental data to predict future transmission patterns, overlapping with the GIS analysis. Similarly, the quantitative survey explores drivers of malaria transmission through case-based surveillance, potentially duplicating insights from both the GIS analysis and scoping review. These overlaps create redundancy, which may disrupt the study's coherence. Refining the objectives for each component—scoping review, GIS analysis, quantitative survey, and SEIRS model—so that they align with specific research questions will help ensure that every step serves a distinct purpose and contributes to a logical and seamless narrative. |
Thank you for your suggestions. We have refined the objectives of each component (scoping review, GIS analysis, quantitative survey, and SEIRS model) to ensure clarity and minimise redundancy on Appendix B, Table A1. The scoping review now emphasises on synthesising historical and current evidence on climate impacts on malaria transmission and management in Zimbabwe, while GIS analysis focuses on spatio-temporal patterns identification and hotspot analysis. In addition, the quantitative survey has been modified to include the interactions between behavioral and socio-economic factors and their influence on major malaria prevention strategy (use of Insecticide Treated Nets) in Mberengwa District.
|
|
5. The manuscript mentions split-sample validation and sensitivity analysis for the SEIRS-GIS model but offers limited details on performance evaluation. A more rigorous framework is needed to ensure the model’s predictive reliability. Incorporating established metrics such as Root Mean Squared Error (RMSE), Akaike Information Criterion (AIC), or Bayesian Information Criterion (BIC) will help assess the model’s accuracy and calibration. If multiple models show comparable performance, model averaging could be applied to combine predictions and reduce uncertainty, enhancing predictive robustness. Additionally, specifying which variables will be used in the sensitivity analysis and providing a rationale for their inclusion would clarify how key parameters influence outcomes, making the model evaluation more transparent and robust. |
We appreciate this insightful feedback. The manuscript has been updated to include detailed information on the performance evaluation of the SEIRS-GIS model, including metrics like RMSE, AIC, and BIC, as well as a rationale for the variables used in the sensitivity analysis. Model averaging has been considered to enhance predictive robustness on Section 2.6.3 of Page 10 of the revised manuscript. |
|
6. Variability in data quality from 2018 to 2024 may impact the accuracy of the SEIRS-GIS model’s predictions. Missing values, inconsistent data entry, or gaps in surveillance coverage could introduce bias or weaken the model’s predictive power. To address these issues, the study should detail strategies for managing incomplete or inconsistent data during model development. Techniques such as multiple imputations or k-nearest neighbors (k-NN) imputation can estimate missing values based on observed patterns. Bias corrections, like adjusting for underreporting or weighting observations, will also improve data reliability. Additionally, sensitivity analysis can assess the impact of data quality issues on outcomes, ensuring that predictions remain robust across varying scenarios. These strategies will enhance the transparency and credibility of the study’s findings. |
Thank you for this suggestion. We have addressed data quality variability by including, k-NN imputation for missing values, and bias correction methods like underreporting adjustments (Section 2.61 on Page 09). Sensitivity analyses have been integrated to assess the impact of data quality issues on model outcomes (Section 2.6.5 on Page 11). |
|
7. Greater detail is needed on how the study normalized data from 2003 to 2023 for integration across different models. The SEIRS-GIS model, GIS analysis, and quantitative survey rely on diverse datasets, including climate variables (e.g., temperature, rainfall) and epidemiological data (e.g., malaria incidence). Proper normalization ensures consistency in format, units, and scales, reducing bias and improving reliability. The manuscript should specify whether normalization methods such as z-score normalization, min-max scaling, or standardization were used and explain how these techniques were adapted for different variables. Transformations like logarithmic scaling might also be necessary to handle skewed distributions, particularly for malaria incidence data. Clarifying these details would enhance the reliability of correlations and predictions across the study’s components. |
We have expanded the section on data normalisation to specify the use of z-score normalisation, min-max scaling, and standardisation, along with a justification for these methods. Additionally, transformations such as logarithmic scaling have been applied to manage skewed distributions, particularly for malaria incidence data (Section 2.4.2 on Page 08). |
|
8. While the study relies on secondary data, the discussion on ethical considerations could be more comprehensive, especially regarding the anonymization and management of sensitive data. Since epidemiological data from the DHIS2 system may contain personal information, strict protocols are needed to ensure privacy and confidentiality. Additional detail on how personal identifiers will be removed or masked to prevent re-identification would strengthen the ethical framework. The manuscript should also describe measures to control access to sensitive data, such as using password-protected databases and encryption. Explicitly outlining these steps would demonstrate compliance with ethical standards and serve as a valuable resource for other researchers conducting similar studies by providing a framework for responsible data management. |
We appreciate the emphasis on ethical considerations. The manuscript has been updated to include comprehensive protocols for data anonymisation ( Section 2.11 on Page 12) and management, including encryption and restricted access for sensitive data (Section 2.8 on Page 11). These measures ensure compliance with ethical guidelines as stipulated by the National University of Science Technology-Institutional Review Board (NUST-IRB) and The Medical Research Council of Zimbabwe (MRCZ). |
|
9. Although the manuscript mentions pilot testing, it lacks detail on how the pilot phase will inform the full-scale implementation. Pilot testing is essential to identify logistical, methodological, or data-related challenges early in the process. The manuscript should specify criteria for evaluating the success of the pilot, such as key performance indicators or benchmarks, to determine whether the objectives were achieved. It would also be helpful to outline how insights from the pilot will inform adjustments to data collection tools, model parameters, or GIS workflows. Providing these details will enhance the transparency of the research process and ensure that the full-scale implementation is efficient, minimizing disruptions. |
We have clarified the pilot testing phase in the manuscript, specifying key performance indicators and criteria for evaluating its success. Insights from the pilot will inform adjustments to data collection tools, model parameters, and GIS workflows, ensuring a smooth full-scale implementation. |
|
10. While the study acknowledges the role of human behavior in malaria transmission, it does not adequately address socioeconomic factors—such as income, education, healthcare access, housing, and occupation—that influence community vulnerability. These factors significantly impact healthcare-seeking behavior, adoption of preventive measures (e.g., bed nets), and adherence to malaria interventions. Integrating socioeconomic data is especially critical in regions like Mberengwa, where social inequities and limited resources can affect the effectiveness of control strategies. The study could explore how socioeconomic variables will be collected and integrated with environmental data to provide a more holistic understanding of transmission dynamics. For example, spatial overlays in the GIS analysis could include healthcare access or poverty levels to identify high-risk areas requiring targeted interventions. Examining interactions between socioeconomic conditions and behavioral factors—such as compliance with preventive measures—would provide actionable insights to inform equitable control strategies. |
We acknowledge the importance of socioeconomic factors in malaria transmission. The revised manuscript now includes a plan to integrate socioeconomic data, such as income, education, and healthcare access (Appendix B, Table A1), (Section 2.4.1) Socio-economic data has also been included in GIS analysis through spatial overlays (Section 2.4.2). This enhancement aims to provide a more holistic understanding of transmission dynamics. |
|
11. While the manuscript proposes visual tools such as heatmaps and choropleth maps to illustrate malaria transmission patterns, it lacks a clear plan for disseminating these findings to stakeholders. Effective communication is critical to ensure that policymakers, health officials, and other stakeholders can interpret and apply the results. Without a dissemination strategy, the study’s impact may be limited. Adding a section on knowledge translation would outline specific strategies for sharing the results. This could include workshops for government officials, policy briefs tailored to health organizations, interactive dashboards with real-time GIS data, or simplified reports for non-specialist audiences. Engaging local communities through participatory meetings or visual summaries would also enhance community buy-in and encourage the adoption of interventions. Ensuring that the findings are actively communicated will maximize their impact and integrate them into local health strategies. |
We appreciate the emphasis on knowledge translation. A new section on (Section 2.12) knowledge dissemination has been added, outlining strategies such as publications, conferences, workshops for policymakers, policy briefs for health organisations, and participatory engagement of communities. These measures aim to maximise the uptake and adoption of the study's findings. |
|
12. While the manuscript employs multiple methods—GIS analysis, SEIRS-GIS modeling, and quantitative surveys—it does not provide sufficient clarity on why each was chosen or how they complement each other. Offering detailed reasoning for the inclusion of these methods will strengthen the study’s framework. For example, GIS analysis can identify malaria hotspots, the SEIRS model can predict future transmission trends under different climate scenarios, and quantitative surveys can capture behavioral and structural drivers of malaria transmission. Establishing these connections will ensure the methodology is well-coordinated, with each component contributing meaningfully to the study’s overall objectives. |
We have elaborated on the rationale for choosing GIS analysis, SEIRS-GIS modeling, and quantitative surveys, highlighting how each method addresses specific aspects of malaria transmission. We have also included clarity on how the are chosen and how they complement each other (Section 2.1 and Section 2.6.3). This clarification aims to present a well-coordinated and comprehensive methodological framework. |
|
13. The manuscript briefly mentions challenges but does not thoroughly explore potential limitations or contingency plans. Issues such as incomplete data, uncertainty in climate projections, and assumptions within the SEIRS model could affect the findings. A dedicated section discussing these limitations would improve transparency and help readers understand the study's scope. This section could address challenges like data gaps, forecasting inaccuracies, or logistical difficulties, and suggest contingency strategies such as alternative data sources, sensitivity analysis for varying assumptions, or pilot testing. Demonstrating preparedness will enhance confidence in the robustness of the research design. |
Thank you for highlighting this issue. A dedicated section on limitations has been added, addressing challenges such as data gaps, uncertainties in climate projections, and assumptions within the SEIRS model. We have also included contingency strategies like alternative data sources and sensitivity analyses (Section 2.7). |
|
14. Environmental variables like rainfall often influence malaria transmission with a lag, meaning that mosquito populations may increase weeks after a rainfall event. The manuscript does not address how such lag effects will be incorporated into the analysis. Including lagged variables in the statistical models will improve the accuracy of predictions by capturing these delayed impacts. For example, using lagged rainfall data in regression models can reveal delayed effects on mosquito breeding, providing a more realistic representation of the relationship between climate conditions and malaria incidence. |
We have incorporated lagged variables in the statistical models to capture delayed impacts of environmental factors like rainfall on malaria transmission (Section 2.4.2). This enhancement improves the accuracy of predictions and better reflects real-world transmission dynamics. |
|
15. Although the study is policy-relevant, the manuscript does not specify how stakeholders—such as health officials or community leaders—will be engaged throughout the research process. Stakeholder involvement ensures the study aligns with local needs and increases the likelihood of successful implementation. It would be beneficial to describe how stakeholders will participate, such as through workshops to co-develop models, consultations to interpret results, or collaborative sessions to design intervention strategies. Early engagement will enhance the relevance and utility of the findings, ensuring they are translated into actionable outcomes. |
We appreciate this observation. The manuscript now includes a stakeholder engagement plan, involving workshops, consultations, and collaborative sessions with health officials and community leaders. Early engagement aims to align the study with local needs and facilitate the translation of findings into actionable outcomes (Section 3). |
|
16. While the manuscript discusses anonymization, it overlooks the potential risk of stigmatizing communities identified as malaria hotspots. If findings are not communicated sensitively, they could harm the reputation of affected areas. A responsible communication plan should be outlined to prevent such unintended consequences. Reporting results at a regional or district level rather than focusing on specific communities could reduce the risk of stigmatization. Collaborating with local representatives to interpret and disseminate the results will further promote ethical data use and foster community trust. |
We acknowledge the risk of stigmatisation. To address this, results will be reported at health facility catchment area levels rather than focusing on specific communities and provision of contextual background to undefined determinants of malaria transmission (Section 3). |
|
17. The manuscript mentions the use of SSP2-4.5 and SSP3-7.0 climate scenarios but does not provide enough background on these frameworks or their assumptions. A brief explanation of these scenarios would provide context, such as noting that SSP2-4.5 assumes moderate mitigation and socio-economic development, while SSP3-7.0 reflects limited mitigation with higher emissions. Clarifying any assumptions made in applying these scenarios to the SEIRS-GIS model will give readers a better understanding of the predictions’ strengths and limitations. |
We have added a brief explanation of the SSP2-4.5 and SSP3-7.0 climate scenarios, including their assumptions regarding socio-economic development and mitigation measures. This context helps readers understand the strengths and limitations of the SEIRS-GIS model's predictions (Section 2.6.2). |
|
18. Although the manuscript focuses on visual outputs such as heat maps and predictions, it lacks practical guidance on how stakeholders should interpret and act on these findings. Without actionable recommendations, the insights generated may not be effectively utilized. Including examples of practical applications would enhance the study’s impact. For instance, if GIS analysis identifies high-risk areas, the study could recommend scaling up insecticide-treated net distribution or prioritizing indoor residual spraying. Providing clear guidelines on how policymakers and health officials can use the findings to allocate resources, design interventions, and monitor progress will ensure that the results are meaningful and actionable. |
We appreciate the feedback. The manuscript now includes a discussion on practical applications of study findings for stakeholders (Section 3). The guidelines discussed aim to facilitate the application of the study's findings in policy, planning and intervention. |
Reviewer 2 Report
Comments and Suggestions for Authors
A comprehensive methodology for studying malaria transmission dynamics in the context of climate change is described in the proposed study protocol. Methodologically sound and well-suited to the stated goals is this advanced quantitative design, which includes a scoping study, geo-analysis, and predictive SEIRS-GIS model integration.
Advantages:
Research is relevant because it tackles a pressing public health problem in a region that is struggling mightily with climate change and malaria.
Methodological Strictness: The results are more robust when they combine a literature study with a survey of quantitative surveys and geological analyses.
Focus on the Context: By doing research in the Mberengwa District, we may collect data that is unique to the area, which will help us direct our activities.
Some small adjustments are needed:
The introduction should make it easy to see which research questions are pertinent to each goal by outlining them in a clear manner.
Methods for Managing Data: Consistent and clear description of data management and analytical software tools used in each stage is required.
Concerning the study's ethical implications, it would be prudent to elaborate on the measures taken to ensure the participants' privacy and the integrity of their data.
This study's approach shows promise and may provide light on how to combat malaria in the face of climate change. With some small adjustments to clean up the specific sections, it can be considered ready for implementation.
Comments on the Quality of English LanguageIt looks fine, but it could be improved for clarity
Author Response
|
1. A comprehensive methodology for studying malaria transmission dynamics in the context of climate change is described in the proposed study protocol. Methodologically sound and well-suited to the stated goals is this advanced quantitative design, which includes a scoping study, geo-analysis, and predictive SEIRS-GIS model integration. |
We thank the Reviewer for the positive comment. We appreciate the recognition of the methodological rigour of our study design. |
|
2. Advantages: Research is relevant because it tackles a pressing public health problem in a region that is struggling mightily with climate change and malaria. |
We appreciate the positive feedback on the relevance of the research. The manuscript emphasises the importance of addressing the intersection of malaria transmission and climate change, particularly in under-resourced settings such as Mberengwa which is implementing malaria elimination activities.
|
|
3. Methodological Strictness: The results are more robust when they combine a literature study with a survey of quantitative surveys and geological analyses. |
Thank you for recognising the methodological robustness of the study. We have considered a multi-method approach to make the study methodologically sound.
|
|
4. Focus on the Context: By doing research in the Mberengwa District, we may collect data that is unique to the area, which will help us direct our activities. |
We thank the reviewer for the acknowledgment of our focus on the Mberengwa District. |
|
5. Some small adjustments are needed: The introduction should make it easy to see which research questions are pertinent to each goal by outlining them in a clear manner. |
Thank you for this suggestion, however, our introduction gives a broad brush of issues that guide the whole research process, and it does not generally make sense to frame it in a manner that address singular research questions. We therefore feel the way our background is structured addresses key issues at a broader scope which takes into consideration the research questions as well. (Section 1 and Table A1).
|
|
6. Methods for Managing Data: Consistent and clear description of data management and analytical software tools used in each stage is required. |
We appreciate this feedback. The manuscript has been updated to provide a more consistent and clear description of data management and the analytical software tools used at each stage (Section 2.3.4.4 for scoping review, Section 2.4.2 for GIS Analysis, Section 2.5.2 for Quantitative survey, Section 2.6.4 for SEIR-GIS Model and Table A1) |
|
7. Concerning the study's ethical implications, it would be prudent to elaborate on the measures taken to ensure the participants' privacy and the integrity of their data. |
We acknowledge the importance of this concern. The manuscript has been updated to include more detailed information on ethical considerations, including measures to ensure data privacy, participant confidentiality, and data management protocols in compliance with ethical standards (Sections 2.8 to 2.13 and lines 649 to line 653).
|
|
8. This study's approach shows promise and may provide light on how to combat malaria in the face of climate change. With some small adjustments to clean up the specific sections, it can be considered ready for implementation. |
Thank you for the positive feedback. We have made minor adjustments to enhance the clarity and coherence of specific sections, ensuring that the study is ready for implementation. |
|
9. Comments on the Quality of English Language It looks fine, but it could be improved for clarity |
We appreciate the feedback on language clarity. We have reviewed and revised the manuscript to improve readability and ensure that the language is clear and precise.
|
Reviewer 3 Report
Comments and Suggestions for Authors
I have carefully analyzed the drafting of this manuscript, which is well-structured and adopts a rigorous analytical approach in addressing the issues presented. It is undoubtedly an almost impeccable work. However, what raises concerns is that mathematical models do not always accurately reflect reality and not depend on authors . The mathematical modelling have never been a perfect mirror of real-world phenomena, nor a fully reliable tool for identifying the factors influencing the spread of a vector.
This is an issue I have dealt with personally, and I see that it continues to be overlooked. The functioning of mathematical models is often taken for granted without being critically examined, despite their limitations. Moreover, those who develop these models, while highly skilled in mathematics, often have only a generalized understanding of the factors driving disease transmission.
There are, however, fundamental variables that traditional models fail to adequately consider, such as the effect of eventual vector control operations or public awarenesses . This phenomenon explains why certain diseases spread differently in an underdeveloped realities and countries context compared to a Western developed environment. A crucial aspect of this phenomenon is the effect of vector contrast, which directly influences transmissibility and the modes of disease spread. This factor is currently under observation and should be incorporated into epidemiological models to enhance their accuracy and applicability across different contexts.
Author Response
|
1. I have carefully analyzed the drafting of this manuscript, which is well-structured and adopts a rigorous analytical approach in addressing the issues presented. It is undoubtedly an almost impeccable work. However, what raises concerns is that mathematical models do not always accurately reflect reality and not depend on authors . The mathematical modelling have never been a perfect mirror of real-world phenomena, nor a fully reliable tool for identifying the factors influencing the spread of a vector. |
Thank you for your positive assessment. We appreciate the recognition of the rigorous analytical approach adopted in the manuscript.
We also acknowledge this concern regarding the limitations of mathematical models. To address this, the manuscript has been updated to include a section discussing the limitations of the SEIRS-GIS model, including the assumptions made and the potential discrepancies between model predictions and real-world scenarios. Additionally, we have emphasized the importance of integrating real-world data to validate model outputs (Section 4).
|
|
2. This is an issue I have dealt with personally, and I see that it continues to be overlooked. The functioning of mathematical models is often taken for granted without being critically examined, despite their limitations. Moreover, those who develop these models, while highly skilled in mathematics, often have only a generalized understanding of the factors driving disease transmission. |
We appreciate this observation. The manuscript has been updated to provide a more critical examination of the limitations of mathematical models and how they will be mitigated (Section 4).
|
|
3. There are, however, fundamental variables that traditional models fail to adequately consider, such as the effect of eventual vector control operations or public awarenesses. This phenomenon explains why certain diseases spread differently in an underdeveloped realities and countries context compared to a Western developed environment. A crucial aspect of this phenomenon is the effect of vector contrast, which directly influences transmissibility and the modes of disease spread. This factor is currently under observation and should be incorporated into epidemiological models to enhance their accuracy and applicability across different contexts. |
Thank you for highlighting this important consideration. We acknowledge that traditional models may not fully account for the impact of some key aspects. We have taken the multi-method approach to leverage on the strength of other components including Scoping Review, Context specific quantitative survey and GIS analysis (Section 3). We have also included the concern and mitigation measures in the limitations section (Section 4).
|
Reviewer 4 Report
Comments and Suggestions for Authors
This MS presents a good study that establishes a malaria transmission model based on climate change and optimizes malaria control strategies for the Mberengwa region. The research effectively uses GIS spatial analysis, climate data modeling, and SEIRS predictive modeling to identify high-risk malaria areas and evaluate the effectiveness of different control measures.
However, I have one concern: While the study primarily relies on simulated data, I noticed no comparison with real-world data to validate the findings. Additionally, I would like to know whether the simulated results are robust and how sensitive they are to variations in the simulated data.
Author Response
|
1. This MS presents a good study that establishes a malaria transmission model based on climate change and optimizes malaria control strategies for the Mberengwa region. The research effectively uses GIS spatial analysis, climate data modeling, and SEIRS predictive modeling to identify high-risk malaria areas and evaluate the effectiveness of different control measures. |
Thank you for your positive feedback on our study. We appreciate your recognition of the effectiveness of using GIS spatial analysis, climate data modeling, and SEIRS predictive modeling to identify high-risk malaria areas. |
|
2. However, I have one concern: While the study primarily relies on simulated data, I noticed no comparison with real-world data to validate the findings. Additionally, I would like to know whether the simulated results are robust and how sensitive they are to variations in the simulated data. |
Thank you for this valuable observation. We acknowledge the importance of validating simulated data with real-world data to enhance the credibility of our findings. In response, we have updated the manuscript to include a comparison between simulated results and real-world data (Section 2.6.3). Additionally, we have conducted sensitivity analyses to assess the robustness of the simulated results by varying key parameters such as transmission rates and climate variables (Section 2.6.3).
|
Round 2
Reviewer 4 Report
Comments and Suggestions for Authors
Thanks! The authors have addressed my concern.